# The Argoverse Trajectory Retrieval Benchmark

**Eric Zhan**
Caltech
ezhan@caltech.edu

**Jagjeet Singh**
Argo AI
jsingh@argo.ai

**Yisong Yue**
Caltech
yyue@caltech.edu

**Andrew Hartnett**
Argo AI
ahartnett@argo.ai

## Abstract

As tracking data becomes more readily available in many domains such as sports, animal tracking, and autonomous vehicles, so does the need for effective information access and retrieval of those growing datasets. To that end, we develop the Argoverse Trajectory Retrieval Benchmark for contextual trajectory retrieval of driving scenarios. The goal of this task is to find similar trajectories from within a large dataset given a query trajectory. This task is challenging because there are many dimensions of variation in which two trajectories can be similar, such as vehicle kinematics, social causality, and road configurations. To our knowledge, this is the first standardized benchmark for trajectory retrieval of driving scenarios. We also provide an evaluation of baseline approaches based on representation learning and relevance feedback, and highlight several areas for improvement for which machine learning can play a large role in future work.

## 1 Introduction

Behavioral tracking data is growing rapidly in many domains, including sports analytics [9, 44, 41], pedestrian crowds [26, 33, 36], traffic scenes [11, 15, 8], and animal behavior [6, 17, 45]. As behavioral track datasets grow, it becomes increasingly important to develop retrieval systems to organize and access information from the data. In this paper, we focus on traffic scenes collected in contexts involving autonomous vehicles (AVs). AV fleets have gathered millions of miles of such log data [11, 15, 8]; having effective information retrieval systems is important for extracting value from the data and accelerating the development of AV technologies.

An effective retrieval system can impact numerous applications, similar to the ubiquity of use cases that exist for web search systems [7, 29]. One use case that motivates our work is dataset curation. For instance, suppose we found an example of an unusual and rare driving maneuver, such as the one depicted in Figure 1a. We can then use our trajectory retrieval system to obtain similar scenes for many possible downstream tasks that can: 1) reveal a better understanding of how often such maneuvers arise; 2) refine our taxonomy of driving behaviors; 3) construct training data to train forecasting models that can more accurately capture such behavior; or 4) create simulations that include such scenarios for safety-critical testing of rare events. Similar retrieval needs arise in related fields such as sports analytics [41, 14, 51].

When designing retrieval systems, especially when using machine learning, it is important to establish standardized benchmarks. To that end, we present the **Argoverse Trajectory Retrieval Benchmark**, which is, to our knowledge, the first standardized retrieval dataset and task for traffic scenes. Our dataset consists of 2,795 scenarios from the Argoverse Motion Forecasting 1.1 validation set [11]. Each scenario has been augmented with relevance labels for 13 complex retrieval intents pertaining to the focal agent and both its social and map contexts. The remaining 240,000+ training and validation scenarios are available for unsupervised methods. We believe that this dataset and retrieval task will stimulate research in designing retrieval systems for behavioral tracking data.

Submitted to the 35th Conference on Neural Information Processing Systems (NeurIPS 2021) Track on Datasets and Benchmarks. Do not distribute.

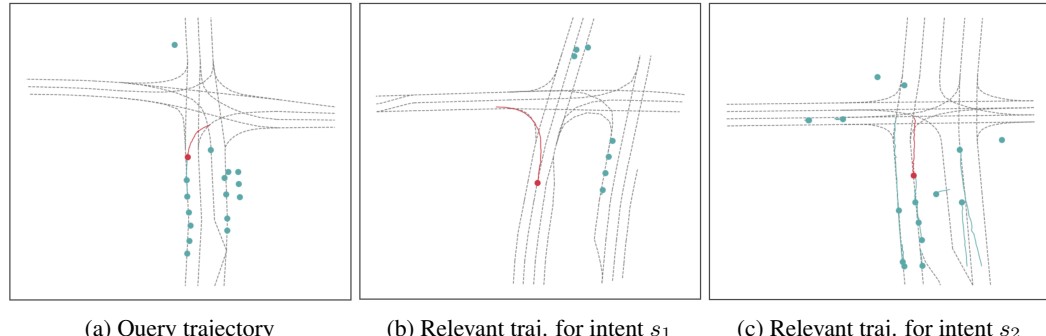

(a) Query trajectory       (b) Relevant traj. for intent $s_1$       (c) Relevant traj. for intent $s_2$

Figure 1: (a) Query trajectory with the focal agent in red. The task is to retrieve trajectories similar to this one. (b) Relevant trajectory for the intent of "turn then change lanes". (c) Relevant trajectory for the intent of "decelerate for moving lead vehicle". Both retrievals are valid for the query, depending on the underlying intent. One challenge with this task is inferring the underlying (hidden) intent.

Benchmarking for traffic scene retrieval poses new design decisions compared to more conventional retrieval domains such as text. The first is defining a suitable similarity measure for retrieval. While such query/item similarity measures are commonly used in retrieval [29], defining such a measure for traffic scenes is challenging. For multi-agent systems like autonomous vehicles, there can be many reasons why two scenes are similar, such as the kinematics of the ego agent (e.g. a significant juke), atypical maneuvers (e.g. k-point turn), an agent's interactions with nearby actors (e.g. yielding to a jaywalker), or the road configuration itself. Importantly, the notion of similarity can differ depending on who is using the retrieval system. For instance, an engineer focused on behavioral understanding and motion prediction for other actors may define similarity by quantifying social influence between actors. Conversely, an engineer focused on motion planner for the AV would likely key in on the maneuver executed by the ego vehicle.

A second challenge is how to model and evaluate longer "information gathering" retrieval sessions that can optionally include relevance feedback. Many of the use cases we have in mind do not fall into the categories for short retrieval sessions such as navigational queries (e.g., the home page of Argo AI) or very specific informational queries (e.g., the number of bridges in Pittsburgh) [7]. For instance in Figure 1, the query trajectory (left) by itself is not enough to distinguish between two possible intents (center, right) without any input from the user. We find in our experiments that catering to multiple definitions of similarity simultaneously without any user feedback is very difficult for a retrieval system, and in fact user feedback may be crucial for developing a practical retrieval system.

To summarize, our contributions are:

1. We present the Argoverse Trajectory Retrieval Benchmark, which enables studying trajectory retrieval for multi-agent systems in a standardized way. We discuss design decisions and provide a benchmark for public use at our dataset page.[1]

2. We establish a suite of baselines, including both hand-crafted feature-based approaches and learned embeddings with state-of-the-art model architectures for AV trajectories.

3. We propose an initial retrieval system that leverages learned embeddings and conduct an evaluation on both the standard and interactive (with relevance feedback) retrieval settings.

4. We conclude with a thorough discussion of our findings and directions for future work.

## 2 Related Work

**Information Retrieval.** Broadly speaking, information retrieval is the study of how to access specific pieces of information within a data repository [29]. The canonical setting is: given a query, retrieve a ranked list of (relevant) results. To date, information retrieval has been studied in many contexts, including web search [7, 29], media retrieval (e.g., music or images) [31, 13, 49], and recently in sports analytics [41, 14, 51]. Sports play retrieval is perhaps the most related to traffic scene

---

[1] https://github.com/ezhan94/argoverse-trajectory-retrieval-benchmark

retrieval, although sports settings tend to be much more structured (fixed number of players, two teams, well-specified objectives, etc.). Furthermore, while some sports trajectory data is publicly available [44] there are currently no standardized retrieval datasets or benchmarks.

Our task is reminiscent of classic retrieval tasks that involve multiple intents or subtopics [32, 39, 55]. In such tasks, two new considerations arose. The first is to be able to "cover" all the different intents or subtopics in order to have some minimal coverage over all intents in a single static ranking [55]. The second is to study interactive ranking settings where users provide so-called relevance feedback [38, 57], after which the retrieval system responds by returning a modified ranking.

**Learning to Rank: Benchmarks & Methods.** Existing benchmarks for information retrieval largely fall under the category of "learning to rank", where there is a set of supervised labels of the form (query, item, relevance level), in addition to a large repository of items [12, 35, 50]. Some datasets may also include information about global query types or genres [3], or query-specific information like intent and subtopics [32]. A related set of benchmarks is based on collaborative filtering, where one is also provided user information [2, 20].

This availability in data has led to significant interest in developing learning algorithms for retrieval (see [27] a broad overview). For retrieval over multiple or ambiguous intents, prior work includes learning for static rankings [54, 42] as well as for dynamic rankings that utilize relevance feedback [5, 52]. These prior work largely use engineered features based on text or metadata (e.g., URL), which can be hard to translate well to our setting. More recent methods that study continuous tracking data typically utilize learned embeddings [47, 21, 51], which we will also use to establish our baselines.

**Trajectory Datasets & Benchmarks.** The rapid growth of AV research opportunities has led to the release of many high-quality large-scale trajectory datasets. These datasets are typically focused on perception issues (detection, segmentation, and tracking) or on issues related to motion forecasting and have widely used benchmarks focused on these tasks. Significant examples include nuScenes [8], the Waymo Open Dataset [48, 15], Lyft Level 5 Dataset [23], and Argoverse [11]. We chose to build our retrieval benchmark on top of the Argoverse Motion Forecasting dataset because it has been widely used by the research community as evidenced by the active leaderboard with more than 225 unique teams as of June 7, 2021. Trajectory benchmarks in other domains include behavior recognition, such as for laboratory animals [6, 17, 45] and human poses [37, 43]. Recent work by Segal et al. [40] is closely related to our proposed benchmark. Segal et al. proposed a method for learning spatio-temporal tags for driving scenes that could then be used for search, and present results on an internal dataset (SDVScenes).

**Trajectory Representation Learning & Modeling.** Modern research on trajectory modeling via representation learning has concentrated on forecasting of future behaviors (e.g., sequential generative modeling) [25, 56, 10, 18, 34], detection of pre-specified behavior categories (e.g., classification) [22, 1, 17], and open-ended knowledge extraction (e.g., unsupervised learning such as clustering) [4, 30, 16]. The study of methods for information access and retrieval of tracking data has received comparatively much less attention, with some exceptions for pose retrieval [47].

## 3  Contextual Multi-Intent Trajectory Retrieval

### 3.1  Problem Description

Let $\tau$ denote a traffic scene trajectory, which can track multiple agents as well as contain contextual information (see Section 4.1). Let $\mathcal{S}$ denote the set of possible intents, i.e. notions of similarity. A *query* is a trajectory-intent pair $(\tau, s), s \in \mathcal{S}$. Our retrieval task is to find and rank trajectories in a retrieval set $\mathcal{R}$ that are similar to $\tau$ with respect to intent $s$. We will denote $\mathcal{Q}$ as the set of queries. The key challenge with our task is that the relevance of a retrieval depends on the intent $s$, but $s$ is hidden from the retrieval system (see Figure 1 for an example). Furthermore, the set of intents $\mathcal{S}$ is also not known ahead of time and can be extended to include new intents in the future.

### 3.2  Quantitative Evaluation

Retrieval systems will be evaluated on how well they rank the trajectories in $\mathcal{R}$ for queries in $\mathcal{Q}$. Let $rel(\tau, s, \tau_q)$ be a scoring function that rates how relevant trajectory $\tau$ is to query $(\tau_q, s)$, with higher scores being more relevant. The ranking metric we use to evaluate a ranked retrieval $\{\tau_1, \ldots, \tau_n\}$ is

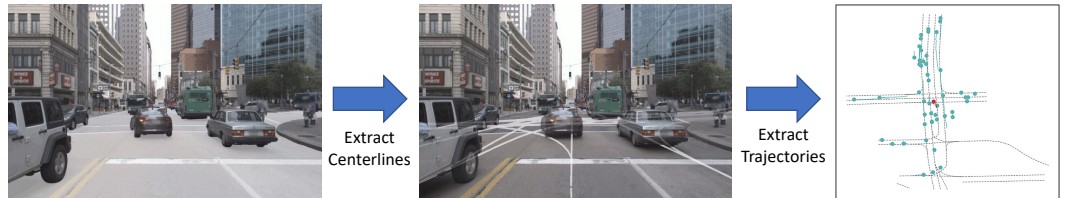

Figure 2: High-level summary of the Argoverse trajectory data format. See [11] for complete details.

the normalized discounted cumulative gain (NDCG):

$$NDCG = \frac{DGC}{iDCG}, \quad DCG = \sum_{i=1}^{n} \frac{rel(\tau_i, s, \tau_q)}{\log_2(i+1)}, \tag{1}$$

where iDCG is computed with respect to the ideal/optimal ranking of $n$ trajectories in $\mathcal{R}$. NDCG is bounded between 0 and 1 is larger for retrievals that rank more relevant trajectories higher. We will compute NDCG and average them over all queries in $\mathcal{Q}$.

### 3.3 Relevance Feedback

Achieving a high NDCG score can be difficult without knowing the hidden intent, as intents can have very different meanings and correspond to different types of trajectories (Figure 1). To address this challenge, we introduce one round of relevance feedback in our benchmark to allow retrieval systems to infer the hidden intent, outlined below:

1. Query $(\tau_q, s)$, retrieval system receives $\tau_q$, $s$ is hidden.
2. Retrieval system returns initial set $\{\tau_1, \ldots, \tau_m\}$.
3. Relevance feedback given to retrieval system $\{rel(\tau_1, s, \tau_q), \ldots, rel(\tau_m, s, , \tau_q)\}$.
4. Retrieval system returns new set $\{\tau_1, \ldots, \tau_n\}$, which can have overlap with the initial set $\{\tau_1, \ldots, \tau_m\}$, and is then scored with NDCG.

These steps simulate a user providing feedback to the retrieval system to allow it to hone in on the hidden intent. In principle, multiple rounds of feedback are possible, but our benchmark will only include one. Instructions for this step will be provided on our dataset page (see appendix).

## 4 The Argoverse Trajectory Retrieval Benchmark

We design our benchmark with the following goals in mind:

1. Multi-intent trajectory retrieval is challenging in domains where data is plentiful, as there can be many dimensions in which two trajectories are similar. To this end, we derive our dataset from the Argoverse Motion Forecasting 1.1 dataset [11], a real-world dataset for trajectory forecasting that contains rich map information with each trajectory (see Figure 2). We describe this process in Section 4.1.

2. Our retrieval task is already very challenging even for simple notions of similarity, so we consider simple intents with scoring functions $rel(\tau, s, \tau_q) = rel(\tau, s)$ to focus on whether or not we're retrieving trajectories for the right intent (the original query trajectory will not affect the score). We describe the labeling process for our intents in Section 4.2. Future iterations of our dataset can consider more complex intents.

3. Lastly, we highlight that the set of intents $\mathcal{S}$ is not fixed. As more data is obtained and annotated (e.g. maps for drivable areas, maps for ground height, etc.), new intents will ultimately be introduced. Ideally, retrieval systems should adapt and be somewhat robust to new intents. To simulate this scenario, we select a subset our intents to only appear in the test query set, described in Section 4.3.

The Argoverse Trajectory Retrieval Benchmark dataset will consist of train/test query sets $\mathcal{Q}_{train}/\mathcal{Q}_{test}$, train/test retrieval sets $\mathcal{R}_{train}/\mathcal{R}_{test}$, the intent set $\mathcal{S}$, and relevance labels $rel(\tau, s)$. We summarize key information about our dataset in Table 1 and Figure 3.

| Intent in $\mathcal{S}$ | All | $\mathcal{Q}_{\text{train}}$ | $\mathcal{Q}_{\text{test}}$ | $\mathcal{R}_{\text{train}}$ | $\mathcal{R}_{\text{train}}$ |
|---|---|---|---|---|---|
| Turn then change lanes | 393 | 18 | 18 | 178 | 179 |
| Straight then turn | 354 | 18 | 19 | 164 | 153 |
| Decelerate then turn | 176 | 10 | 7 | 85 | 74 |
| Turn then decelerate | 133 | 39 | 10 | 38 | 46 |
| Decelerate for stationary LV | 251 | 25 | 10 | 115 | 101 |
| Decelerate for moving LV | 425 | 65 | 8 | 173 | 179 |
| Decelerate to a stop | 610 | 82 | 15 | 260 | 253 |
| Decelerate after intersection | 237 | 76 | 14 | 75 | 72 |
| Test intent #1 | 228 | 0 | 12 | (104) | 112 |
| Test intent #2 | 526 | 0 | 12 | (258) | 256 |
| Test intent #3 | 815 | 0 | 21 | (393) | 401 |
| Test intent #4 | 305 | 0 | 20 | (136) | 149 |
| Test intent #5 | 245 | 0 | 17 | (113) | 115 |
| Total # trajectories | 2,795 | 100 | 50 | 1,323 | 1,322 |
| Avg. # intents/trajectory | 1.68 | 3.33 | 3.66 | 1.58 | 1.58 |
| # trajectory-intent queries | n/a | 321 | 170 | n/a | n/a |

Table 1: # of trajectories with each intent (counted if $rel(\tau, s) > 0$) for all query and retrieval sets. LV = leading vehicle. $\mathcal{Q}_{\text{train}}$ contains no trajectories with test intents while $\mathcal{R}_{\text{train}}$ does, but the labels are not provided. Summary statistics are included in the last 3 rows. We only consider a trajectory-intent pair as a query if $rel(\tau, s) = 2$.

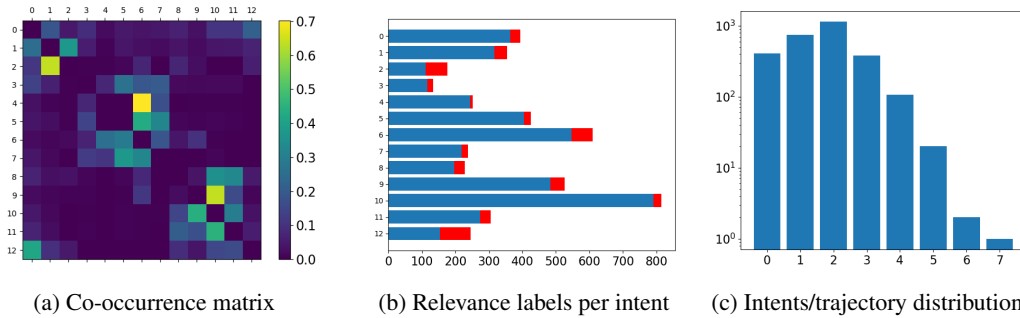

(a) Co-occurrence matrix     (b) Relevance labels per intent     (c) Intents/trajectory distribution

Figure 3: (a) Co-occurrence matrix of all intents of 2,795 trajectories. The matrix is not symmetric because it is row-normalized, i.e. cell $o_{ij}$ is the percentage of trajectories with intent $s_i$ that also have intent $s_j$. The order of intents is same as in Table 1. (b) Counts of relevance 2 (blue) and relevance 1 (red) labels for each intent. The order of intents is same as in Table 1. (c) Distribution of the # of intents per trajectory in log-scale. Trajectories have at most 7 intents in our dataset.

## 4.1 Constructing the Dataset

Our dataset is derived from the Argoverse Motion Forecasting 1.1 dataset [11], which extracts planar trajectories and centerlines from sequences of LiDAR and camera images (see Figure 2). Each trajectory is 5 seconds long and tracks $K > 1$ agents at 10 Hz ($T = 50$). We let $\mathbf{x}_t^k \in \mathbb{R}^2$ denote the $k$-th agent's planar $(x, y)$ coordinates at time $t$. Similarly, denote $\mathbf{X}_t := \{\mathbf{x}_t^1, \dots, \mathbf{x}_t^K\}$, $\mathbf{X}^k := \{\mathbf{x}_1^k, \dots, \mathbf{x}_T^k\}$, and $\mathbf{X} = \{\mathbf{X}_1, \dots, \mathbf{X}_T\} = \{\mathbf{X}_1, \dots, \mathbf{X}_K\}$. $\mathbf{X}^1$ will always denote the focal agent and is visualized in red, while all other agents in teal (see Figure 1). Each trajectory also contains contextual information $\mathbf{C} = \{\mathbf{C}_1, \dots, \mathbf{C}_T\}$, some of which may change over time (e.g. nearest centerline to focal agent) while others remain static (e.g. lane connectivity graph). We refer to the original Argoverse paper [11] for the complete details. In summary, our trajectories $\tau$ consist of tracking information $\mathbf{X}$ and contextual information $\mathbf{C}$: $\tau = (\mathbf{X}, \mathbf{C})$.

We filter the Argoverse Motion Forecasting validation set that initially contains 39,472 trajectories using automatic labeling functions to find "interesting" trajectories that contain more complex maneuvers and/or social interactions. We filter for features such as large acceleration, large deceleration, leading vehicles, traffic control, etc., and refine the validation set down to 2,975 trajectories that we label with intents in Section 4.2. This filtering step will be included with our code release.

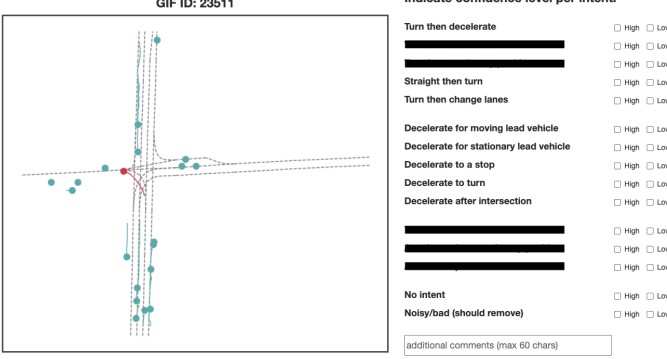

Figure 4: Interface for labeling trajectories. Annotators are shown a video of a trajectory on the left and asked to label the relevance of each intent on the right, including test intents (redacted in this figure). Options are high relevance (2), low relevance (1), or no selection (0). There is an additional option at the bottom to remove a trajectory.

## 4.2 Labeling Trajectories with Intents from $\mathcal{S}$

We focus on intents that describe or lead to complex behaviors, such as intents with an "A THEN B" structure (e.g. turn THEN change lanes, then THEN decelerate) and intents that capture social interaction (e.g. decelerate for leading vehicle). In total, $\mathcal{S}$ contains 13 intents in $\mathcal{S}$, listed in Table 1. We annotate all trajectory-intent pairs by labeling $rel(\tau, s)$ as one of 3 degrees of relevance: $\{0, 1, 2\}$ for $\{$not, somewhat, highly$\}$ relevant. The labeling details are as follows:

1. Initially, 2 domain expert each labeled roughly half of the 2,975 filtered trajectories using the interface depicted in Figure 4. An option to remove a trajectory was available to address issues like trajectory jitter or over-segmentation.

2. Trajectories with more than 2 labeled intents were then labeled again by the other expert.

3. For trajectories that were labeled twice, we considered labels to be in agreement if they were the same, were 0 and 1 (defaulted to 0) or were 1 and 2 (defaulted to 2). Roughly 80% of double-labels were in agreement.

4. For labels that were in disagreement (0 and 2), the domain experts resolved them together.

Label statistics are summarized in Table 1 and visualized in Figure 3. 175 of the initial 2,970 trajectories were ultimately removed, bringing the final total to 2,795 trajectories. 1,144 trajectories were labeled by both experts while 1,651 have a single set of labels. Labeling took 30hrs combined.

## 4.3 Query Selection and Train/Test Split

We first split the 2,795 labeled trajectories into query and retrieval sets. For queries, we manually select 150 trajectories that have multiple intents (at least 2 intents with a high relevance label of 2) and such that the intents have good coverage over the intent set $\mathcal{S}$. The remaining trajectories comprise the retrieval set.

Next, we split $\mathcal{Q}$ and $\mathcal{R}$ into train and test sets such that only 8 of the 13 intents appear in $\mathcal{Q}_{\text{train}}$, while all 13 are represented in $\mathcal{Q}_{\text{test}}$. This simulates the real-world scenario of having to adapt to newly encountered intents. Then we split the retrieval set into $\mathcal{R}_{\text{train}}$ and $\mathcal{R}_{\text{test}}$ such that they have similar distributions over intents. Refer to Table 1 for full details about our query and retrieval sets.

Our dataset will provide all relevance labels $rel(\tau, s)$ for the 8 train intents for trajectories in $\mathcal{Q}_{\text{train}}$ and $\mathcal{R}_{\text{train}}$. Queries in $\mathcal{Q}_{\text{test}}$ will be provided with masked intents, and retrieval systems will be evaluated on how well they rank the trajectories in $\mathcal{R}_{\text{test}}$ via NDCG score.

## 5 Baseline Experiments

Defining a similarity measure between two trajectories can be challenging due to dealing with many modalities (maps, trajectories, logged metadata, etc.). Traditional methods that rely on feature engineering and feature matching may have trouble scaling as more data is collected. Recent work has instead focused on learning embedding functions that encode input data into a lower-dimensional

vector (or embedding) space. The advantage is that similarity can be more intuitively understood as distance in embedding space, but we lose the ability to interpret what information is retained in the embeddings. Nevertheless, learning trajectory embeddings have been shown to be effective for many downstream tasks [24, 18, 46], and so we establish our retrieval baselines in this way. Our main evaluation results are described Table 2, which we will discuss throughout this section.

## 5.1 Retrieval via Nearest Neighbors in Embedding Space

We define an embedding function as $\mathbf{f}_\theta$ parameterized by $\theta$ that encodes a trajectory $\tau$ into a lower-dimensional embedding vector $\mathbf{z} = \mathbf{f}_\theta(\tau)$. The information retained in the embedding ultimately depends on the auxiliary task used to train the model (e.g. forecasting vs. autoencoding). $\mathbf{f}_\theta$ itself can take on many forms and contain

---
**Algorithm 1** Nearest-Neighbor$((\tau_q, s), \mathcal{R}, n, \mathbf{d}(\cdot), \mathbf{f}_\theta)$

---
1: **Inputs**: query $(\tau_q, s)$, retrieval set $\mathcal{R}$, top-$n$ trajectories
2: **Inputs**: distance function $\mathbf{d}(\cdot)$, embedding function $\mathbf{f}_\theta$
3: Compute query embedding $\mathbf{z_q} = \mathbf{f}_\theta(\tau_\mathbf{q})$.
4: Compute embeddings $\mathbf{z_i} = \mathbf{f}_\theta(\tau_\mathbf{i})$ for $\tau_i \in \mathcal{R}$.
5: Rank $\tau_i \in \mathcal{R}$ in increasing order of $\mathbf{d}(\mathbf{z_i}, \mathbf{z_q})$.
6: **Output**: top-$n$ closest trajectories to query

---

multiple components, such as hand-crafted features, sliding window operations [53], recurrent neural networks [24], and graph attention networks for capturing social interactions [18].

Given an embedding function $\mathbf{f}_\theta$, we can design a ranked retrieval system that returns the top-$n$ trajectories in the retrieval set with embeddings closest to the query embedding with respect to some distance function (a common choice is Euclidean distance: $\mathbf{d}_{\text{Euclidean}}(\mathbf{z_i}, \mathbf{z_j}) = \|\mathbf{z_i} - \mathbf{z_j}\|_\mathbf{2}$, [41]). This algorithm is outlined in Algorithm 1 and has time complexity $O(|\mathcal{R}| \log n)$ per query if implemented with a heap. Note that the algorithm does not take into account a hidden intent and always returns the same retrievals for each query trajectory.

We consider 4 embeddings functions $\mathbf{f}_\theta$, described below:

1. **FEAT** - a naive embedding function that simply computes 15 domain-specific features, such as average speed of the focal agent, the curvature of its trajectory, and its distances to other agents. There are no parameters to be learned for this embedding function.

2. **AE** - a simple autoencoder for the focal agent trajectory $\mathbf{X^1}$ implemented with a recurrent neural network for both the encoder and decoder. Other agents $\{\mathbf{X^2}, \dots, \mathbf{X^K}\}$ and contextual information $\mathbf{C}$ are ignored.

3. **WIMP** [24] - a state-of-the-art model for trajectory forecasting that encodes all agents as well as the nearest centerline to the focal agent. We use the same model architecture but train it to reconstruct the focal agent trajectory $\mathbf{X^1}$.

4. **VNET** [18] - VectorNet, another state-of-the-art model trained for trajectory forecasting (forecast next 3sec given a 2sec history) that includes a graph attention network for encoding all contextual map information and also a node reconstruction task in its objective. We use the node embedding for the focal agent full 5sec trajectory.

The embedding functions are developed using the Argoverse Motion Forecasting 1.1 training set. We evaluate nearest-neighbor retrieval using NDCG and the query and retrieval sets constructed in Section 4.3 and report our results in Table 2 (rows with $m = 0$, "standard" columns). We observe that NDCG decreases as the number of retrievals $n$ increases because retrieving a larger optimal set is more difficult. Out of all the embeddings, WIMP performs the best. We hypothesize that this is the case because WIMP is trained to reconstruct the focal agent trajectory and all queries pertain to said focal agent. On the other hand, VectorNet is trained for trajectory forecasting and performs the worst. We reason that this occurs because VectorNet embeddings must retain some information about possible futures (and also information for node completion), which can be irrelevant for comparing embeddings of trajectory histories. We note that the hand-crafted FEAT embedding performs reasonably well, although noticeably worse than the best learned embedding. We conclude that embeddings trained for trajectory reconstruction are better suited for our retrieval task.

## 5.2 Triplet Loss Fine-tuning with $\mathcal{Q}_{\textbf{train}}$, $\mathcal{R}_{\textbf{train}}$

In our next set of experiments, we use the relevance labels given in $\mathcal{Q}_{\text{train}}$ and $\mathcal{R}_{\text{train}}$ to fine-tune embeddings with a triplet loss. Our motivation is that having trajectories with the same intent labels

| Query Set | NDCG | | standard (Section 5.1) | | | | triplet fine-tuning (Section 5.2) | | | |
|---|---|---|---|---|---|---|---|---|---|---|
| | $n$ | $m$ | FEAT | AE | WIMP | VNET | FEAT | AE | WIMP | VNET |
| $\mathcal{Q}_{\text{train}}$ | 10 | 0 | .379 | .349 | .391 | .230 | .385 | .370 | .385 | .243 |
| | 30 | 0 | .352 | .334 | .374 | .231 | .367 | .359 | .376 | .238 |
| | 50 | 0 | .345 | .330 | .368 | .231 | .361 | .358 | .371 | .236 |
| $\mathcal{Q}_{\text{train}}$ | 10 | 5 | .411 | .425 | .410 | .236 | .399 | .429 | .414 | .256 |
| | 30 | 5 | .365 | .367 | .366 | .209 | .352 | .377 | .375 | .219 |
| | 50 | 5 | .343 | .346 | .348 | .203 | .336 | .361 | .360 | .203 |
| $\mathcal{Q}_{\text{test}}$ | 10 | 0 | .337 | .355 | .371 | .273 | .334 | .331 | .355 | .273 |
| | 30 | 0 | .310 | .324 | .343 | .261 | .318 | .318 | .336 | .257 |
| | 50 | 0 | .305 | .310 | .328 | .254 | .311 | .313 | .331 | .250 |
| $\mathcal{Q}_{\text{test}}$ | 10 | 5 | .409 | .429 | .436 | .236 | .378 | .394 | .396 | .272 |
| | 30 | 5 | .353 | .373 | .390 | .208 | .339 | .359 | .365 | .246 |
| | 50 | 5 | .332 | .343 | .367 | .202 | .324 | .343 | .351 | .238 |

Table 2: NDCG scores for queries in $\mathcal{Q}_{\text{train}}, \mathcal{Q}_{\text{test}}$ and retrievals from $\mathcal{R}_{\text{train}}, \mathcal{R}_{\text{test}}$ respectively. $n$ is the # of retrievals, $m$ is the # of trajectories for relevance feedback. 1) NDCG decreases as $n$ increases, as retrieving a larger optimal set is more difficult. 2) Utilizing relevance feedback leads to clear improvement for all embeddings except VNET. 3) Triplet fine-tuning does *not* lead a clear improvement. 4) There is generally not a big difference in performance between train and test queries, but the difference is larger for fine-tuned embeddings, possibly because of overfitting to training intents. 5) Overall, WIMP embeddings without fine-tuning appear to be the best for our retrieval task.

closer together in embedding space will improve nearest neighbor retrieval.[2] In particular, we train an autoencoder $(\mathbf{g}_{\text{enc}}, \mathbf{g}_{\text{dec}})$ that minimizes the following objective:

$$\underbrace{\max(\|\mathbf{g}_{\text{enc}}(\mathbf{z}) - \mathbf{g}_{\text{enc}}(\mathbf{z}_{\text{pos}})\|_2 - \|\mathbf{g}_{\text{enc}}(\mathbf{z}) - \mathbf{g}_{\text{enc}}(\mathbf{z}_{\text{neg}})\|_2 + \alpha, 0)}_{\text{triplet loss}} + \underbrace{\|\mathbf{z} - \mathbf{g}_{\text{dec}}(\mathbf{g}_{\text{enc}}(\mathbf{z}))\|_2}_{\text{reconstruction loss}}. \quad (2)$$

$(\mathbf{z}, \mathbf{z}_{\text{pos}}, \mathbf{z}_{\text{neg}})$ is a triplet of embeddings where $\mathbf{z}, \mathbf{z}_{\text{pos}}$ share the same label while $\mathbf{z}, \mathbf{z}_{\text{neg}}$ do not. The triplet loss in (2) encourages embeddings with the same label to be closer together than embeddings with different labels, up to some margin $\alpha$. At the same time, we aim to retain the same information encoded in the original embeddings by including the standard autoencoder reconstruction loss in (2).

We construct triplets $(\mathbf{z}, \mathbf{z}_{\text{pos}}, \mathbf{z}_{\text{neg}})$ by considering every trajectory-intent pair $(\tau, s)$ with $rel(\tau, s) = 2$ in $\mathcal{Q}_{\text{train}} \bigcup \mathcal{R}_{\text{train}}$. For each pair, we sample a positive trajectory $\tau_{\text{pos}}$ from those in $\mathcal{Q}_{\text{train}} \bigcup \mathcal{R}_{\text{train}}$ that share the same label ($rel(\tau_{\text{pos}}, s) = 2$), and similary we sample a negative trajectory ($rel(\tau_{\text{neg}}, s) = 0$). Triplets are re-sampled at the beginning of every epoch (e.g. offline triplet mining).

The new embedding function we use for our retrieval system is then $\mathbf{z} = \mathbf{g}_{\text{enc}}(\mathbf{f}_\theta(\tau))$ and we report our results in Table 2 ("triplet fine-tuning" columns). We observe that the results are inconsistent: NDCG can both increase/decrease compared to the "standard" embedding columns. Furthermore, we see that there is generally a drop in performance on the query test set, which likely occurs because the query test set contains test intents that were not fine-tuned with our triplet loss. We note that our fine-tuning step is applied after training the initial embeddings so there might be some information loss (that we tried to mitigate with the autoencoding loss in (2)). Future work should consider jointly training embeddings with the triplet loss.

## 5.3 Retrieval with Relevance Feedback

Our previous two experiments ignore a main challenge of our problem setting by disregarding that there is a hidden intent and will always return the same set of trajectories for each query. In our final experiment, we design a retrieval system that utilizes the relevance feedback procedure described in Section 3.3 to address this challenge. We consider a version of nearest neighbor retrieval in Algorithm 1 that uses an updated distance function given the relevance feedback, as described in Algorithm 2.

Let $(\tau_q, s)$ be our initial query and $\mathcal{M} = \{\tau_1, \ldots, \tau_m\}$ be our initial set of $m$ retrievals for which we receive relevance feedback $\{rel(\tau_1, s), \ldots, rel(\tau_m, s)\}$. We construct two sets: relevant set

---

[2]Indeed, triplet or contrast loss has been used in other related retrieval settings, such as for human poses [47].

$\mathcal{A} = \{\tau | rel(\tau, s) > 0, \tau \in \mathcal{M}\} \bigcup \{\tau_q\}$ and non-relevant set $\mathcal{B} = \{\tau | rel(\tau, s) = 0, \tau \in \mathcal{M}\}$. We consider an updated distance function that prioritizes trajectories with embeddings close to the relevant set and far from the non-relevant set:

$$\mathbf{d}_{\mathcal{AB}}(\mathbf{z}, \mathbf{z_q}) = \frac{1}{|\mathcal{A}|} \sum_{\tau \in \mathcal{A}} \mathbf{d}(\mathbf{z}, \mathbf{f}_\theta(\tau)) - \frac{1}{|\mathcal{B}|} \sum_{\tau \in \mathcal{B}} \mathbf{d}(\mathbf{z}, \mathbf{f}_\theta(\tau)). \tag{3}$$

(3) is reminiscent of the Rocchio algorithm [28] except we compute the distances to the relevant set rather than update the query embedding directly. Algorithm 2 summarizes our approach incorporating relevance feedback and has time complexity $O(|\mathcal{R}|(\log n + \log m))$ per query.

We observe in Table 2 (rows with $m = 5$) that leveraging relevance feedback improves NDCG for all embeddings (except VectorNet). This matches our intuition because our approach in Algorithm 2 uses feedback given by the simulated user to refine the retrieval for the hidden intent. These results suggest user feedback, even a limited amount, can be crucial for efficient multi-intent trajectory retrieval.

---

**Algorithm 2** Nearest-Neighbor-with-Relevance-Feedback$((\tau_q, s), \mathcal{R}, n, m, \mathbf{d}(\cdot), \mathbf{f}_\theta)$

---

1: **Inputs**: query $(\tau_q, s)$, retrieval set $\mathcal{R}$, top-$n$ trajectories, $m$ feedback
2: **Inputs**: distance function $\mathbf{d}(\cdot)$, embedding function $\mathbf{f}_\theta$
3: $\mathcal{M}$ = Nearest-Neighbor$((\tau_q, s), \mathcal{R}, m, \mathbf{d}(\cdot), \mathbf{f}_\theta)$ using Algorithm 1.
4: Receive relevance feedback for trajectories in $\mathcal{M}$.
5: Construct sets $\mathcal{A}$, $\mathcal{B}$, and update distance function $\mathbf{d}_{\mathcal{AB}}$ in (3).
6: **Output**: Nearest-Neighbor$((\tau_q, s), \mathcal{R}, n, \mathbf{d}_{\mathcal{AB}}(\cdot), \mathbf{f}_\theta)$ using Algorithm 1.

---

# 6 Discussion and Future Work

We have introduced the Argoverse Trajectory Retrieval Benchmark for standardizing the challenging task of multi-intent retrieval in the domain of AV trajectories. We explore initial baseline retrieval algorithms that use trajectory embeddings and summarize our findings:

1. Embeddings trained to reconstruct rather than forecast the focal agent trajectory are better-suited for queries that pertain to the focal agent (Section 5.1).

2. Triplet loss fine-tuning with relevance labels does not appear to be effective, but a joint training approach has yet to be explored (Section 5.2).

3. Incorporating relevance feedback may be key for this retrieval setting (Section 5.3).

Our benchmark is the first iteration of what we expect to be a promising research area. There are many directions for future work and many more challenges to overcome as we continue to scale up. For instance, the trajectory data provided in Argoverse is only a small subset of the data that's available, such as richer map information like ground height, agent type (vehicle vs. pedestrian), and the state of traffic control. As more data is incorporated, intents will grow in number and complexity and retrieval systems may fail to scale accordingly. Another direction for future work is to consider more diverse queries beyond those that pertain to the focal agent, as embeddings trained to reconstruct the focal agent trajectory is unlikely to be the best solution for all query types. Potential solutions may use multiple embeddings trained with different auxiliary tasks within their retrieval systems. A third direction is explore other forms of relevance feedback, such as pairwise comparisons or ranking an initial retrieval set. It is unclear what form of relevance feedback is the most informative for retrieval systems and also easy for users to provide.

Ultimately, further progress in this research direction will come from scaling up our benchmark. For instance, approaches may overfit to our set of intents that all pertain to the focal agent. We try to prevent this by having held-out test intents, and we also expect future versions of our dataset to include more diverse queries. Lastly, it's important to understand that the usefulness of retrieval systems is tied to the underlying data and can be subject to biases of the data. Thus, some scenarios may intrinsically be harder to retrieve than others. Diagnosing biases in retrieval systems could be another interesting direction for future work.

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

## A    Key Information

**Dataset page**: `https://github.com/ezhan94/argoverse-trajectory-retrieval-benchmark`.
All relevant information can be found at our dataset page linked above (dataset download, code, license, instructions for submitting a benchmark, additional supplementary materials, etc.)

**Dataset documentation and intended uses**: we use the datasheets for datasets framework [19] in Appendix B.

**Author statement**: We bear all responsibility in case of violation of rights, etc., and confirmation of the data license.

**Hosting, licensing, and maintenance plan**: This information will be provided on our dataset page.

## B    Datasheets for Datasets [19]

### B.1    Motivation

- **For what purpose was the dataset created?**  The task of finding "similar" scenes or trajectories within a large corpus of log data has proven challenging. Existing "learning to rank" systems do not readily port to this trajectory domain. This dataset was created to enable and encourage further research on trajectory retrieval in the setting of AV development.

- **Who created the dataset (e.g., which team, research group) and on behalf of which entity (e.g., company, institution, organization)?**  The prediction team at Argo AI in collaboration with Caltech.

- **Who funded the creation of the dataset?** Argo AI.

- **Any other comments?** None.

### B.2    Composition

- **What do the instances that comprise the dataset represent (e.g., documents, photos, people, countries)?**  In this work we add relevance labels to a subset scenarios of the Argoverse Motion Forecasting validation set. The underlying scenarios represent the planar centroid positions of actors in a traffic scene. Each 5s (10Hz) scenario has been derived from a AV log and contains at least one agent that is present for the entire 5s and performs a significant action.

- **How many instances are there in total (of each type, if appropriate)?.**  We add 13 relevence labels to 2,795 scenarios. Label statistics are shown in Table 1 and Figure 3.

- **Does the dataset contain all possible instances or is it a sample (not necessarily random) of instances from a larger set?**. We label 2,795 of the 39,472 scenarios comprising the Argoverse 1.1 Motion Forecasting validation set. 2,970 scenarios were selected using automatic labeling functions to find "interesting" trajectories that contain more complex maneuvers or social interactions. We detect features like the presence of acceleration, deceleration, leading vehicles, traffic control, etc. 175 scenarios were removed during labeling. These scenarios eliminated for tracking errors such as id-swaps or over-segmentation of the focal track.

- **What data does each instance consist of?**  Each Argoverse scenario consists of planar centroid positions for actors in a traffic scene. These centroids are sampled at 10Hz and the full duration of the scene is 5s. A lane graph and underlying lane centerlines are also provided. Here we add relevance labels $\in \{0, 1, 2\}$ for each of 13 intents to each selected scenario.

- **Is there a label or target associated with each instance?**. For each of 2,795 there are 13 relevance labels associated with the underlying scenario.

- **Is any information missing from individual instances?** Relevance labels corresponding to 5 of the 13 intents are hidden for all training examples. All test set labels are also hidden.

- **Are relationships between individual instances made explicit (e.g., users' movie ratings, social network links)?** Not applicable.

- **Are there recommended data splits (e.g., training, development/validation, testing)?** Yes. Items are split into test queries, test retrievals

- **Are there any errors, sources of noise, or redundancies in the dataset?** The underlying Argoverse scenarios represent a real urban driving dataset; there is an expected degree of tracking noise and segmentation errors. Relevance labels were provided by domain experts but nevertheless may contain noise due to human error or subjective judgement.

- **Is the dataset self-contained, or does it link to or otherwise rely on external resources (e.g., websites, tweets, other datasets)?** The retrieval benchmark is built upon another existing dataset. However, the retrieval benchmark labels will be hosted with the requisite forecasting scenarios.

- **Does the dataset contain data that might be considered confidential (e.g., data that is protected by legal privilege or by doctorpatient confidentiality, data that includes the content of individuals' non-public communications)?** No.

- **Does the dataset contain data that, if viewed directly, might be offensive, insulting, threatening, or might otherwise cause anxiety?** No.

- **Does the dataset relate to people?** No.

### B.3 Collection Process

- **How was the data associated with each instance acquired?**. The underlying Argoverse Motion Forecasting scenarios were captured by an AV (part of the Argo AI fleet). Each AV is equipped with multiple cameras, lidar, and radar. Raw sensor data is processed to produce tracks localized on a pre-constructed map. Full details are available in [11]. The relevance labels provided in this work were provided by two domain expert labelers using the labeling tool depicted in Figure 4.

- **What mechanisms or procedures were used to collect the data (e.g., hardware apparatus or sensor, manual human curation, software program, software API)?** Scenarios were selected through hand crafted labeling functions. Relevence scores were added using the web-app labeling tool shown in Figure 4.

- **If the dataset is a sample from a larger set, what was the sampling strategy (e.g., deterministic, probabilistic with specific sampling probabilities)?**. Scenarios for labeling were chosen using a set of labeling functions designed to identify complex and interesting scenarios.

- **Who was involved in the data collection process (e.g., students, crowdworkers, contractors) and how were they compensated (e.g., how much were crowdworkers paid)?** Data was collected by employees of Argo AI.

- **Over what timeframe was the data collected?** Source logs Argoverse scenarios were collected over several months in 2019.

- **Were any ethical review processes conducted (e.g., by an institutional review board)?** Not applicable to the relevance labels outlined in this work.

- **Does the dataset relate to people?** No.

### B.4 Preprocessing/cleaning/labeling

- **Was any preprocessing/cleaning/labeling of the data done (e.g., discretization or bucketing, tokenization, part-of-speech tagging, SIFT feature extraction, removal of instances, processing of missing values)?** 175 of the programmatically selected scenarios were excluded at the discretion of the labelers. Additionally, automated procedures were used to resolve a significant set of slightly disparate results across labelers. See section 4.2 for details.

- **Was the "raw" data saved in addition to the preprocessed/cleaned/labeled data (e.g., to support unanticipated future uses)?** All raw labels were saved but are not part of the publicly released benchmark.

- **Is the software used to preprocess/clean/label the instances available?** No. These are simple heuristics outlined in section 4.2.

- **Any other comments?** None.

## B.5 Uses

- **Has the dataset been used for any tasks already?** The underlying scenarios from Argoverse 1.1 have been used extensively for tracking and motion forecasting competitions. The new relevance labels for the retrieval task have not been used outside of the presented baselines.

- **Is there a repository that links to any or all papers or systems that use the dataset?** Not applicable.

- **What (other) tasks could the dataset be used for?** Our intent labels can also be used as the first step towards establishing a taxonomy of driving behaviors.

- **Is there anything about the composition of the dataset or the way it was collected and preprocessed/cleaned/labeled that might impact future uses?** Our dataset does not include full contextual information (e.g. camera images and 3D shapes), which impacts what conclusions we can draw about this dataset.

- **Are there tasks for which the dataset should not be used?** No. **Any other comments?** None.

## B.6 Distribution

- **Will the dataset be distributed to third parties outside of the entity (e.g., company, institution organization) on behalf of which the dataset was created?** The benchmark will be publicly available under a non-commercial license.

- **How will the dataset will be distributed (e.g., tarball on website, API, GitHub)?** Data will be availble for tarball download through the existing Argoverse website. Test labels are hidden and test performance can only be obtained via API calls to an evaluation server.

- **When will the dataset be distributed?** Our current plan is to publicly release the dataset by July 1, 2021 on our dataset page.

- **Will the dataset be distributed under a copyright or other intellectual property (IP) license, and/or under applicable terms of use (ToU)?** We intend to release the data under the Creative Commons Attribution-NonCommercial-ShareAlike 4.0 International Public License ("CC BY-NC-SA 4.0"). Terms of use for all Argoverse data are posted at `https://www.argoverse.org/about.html#terms-of-use`

- **Have any third parties imposed IP-based or other restrictions on the data associated with the instances?** No.

- **Do any export controls or other regulatory restrictions apply to the dataset or to individual instances?** No.

- **Any other comments?** None.

## B.7 Maintenance

- **Who is supporting/hosting/maintaining the dataset?** Data will be supported and hosted as part of the Argoverse project by Argo AI.

- **How can the owner/curator/manager of the dataset be contacted (e.g., email address)?** Dataset owners can be contacted via email (ahartnett@argo.ai or ezhan@caltech.edu) or via github issues at `https://github.com/argoai/argoverse-api/issues`

- **Is there an erratum?** Not currently, though one can be added if errors are discovered.

- **Will the dataset be updated (e.g., to correct labeling errors, add new instances, delete instances)?** Yes and the version number will be incremented.

- **If the dataset relates to people, are there applicable limits on the retention of the data associated with the instances (e.g., were individuals in question told that their data would be retained for a fixed period of time and then deleted)?** Not applicable.

- **Will older versions of the dataset continue to be supported/hosted/maintained?** Deprecated version of the dataset will be hosted but labeled as deprecated.

- **If others want to extend/augment/build on/contribute to the dataset, is there a mechanism for them to do so?** Please reach out via `https://github.com/argoai/argoverse-api/issues`.
- **Any other comments?** None.

