# OpenReview forum: "The Argoverse Trajectory Retrieval Benchmark"
_NeurIPS.cc/2021/Track/Datasets_and_Benchmarks/Round1 — Submitted to NeurIPS 2021 Datasets and Benchmarks Track (Round 1)_

### Official Review · Reviewer_FmPW · 2021-07-03
**Initial review of the Argoverse Benchmark**

**Rating:** 4
**Confidence:** 2
**Clarity:** The paper is fairly well written and …

**Strengths:**

It is claimed to be the first work on standardized benchmark for trajectory retrieval under driving scenario.

The authors provide a number of baseline evaluation results based on diverse set of features/methods.

**Weaknesses:**

It is not clear whether an additional explicit intent label is required to capture interesting information arising from a social interaction. For one, one of the experiments, FEAT, which only uses domain specific hand crafted features without an intent, seems to do comparably to a more elaborate one, WIMP. In fact, there is no strong evidence in paper text that intent was used in any of the experiments in section 5.1.

Second, if intent were used meaningfully to contribute to improvements ranking, there should be an ablation study that corroborates its utility.

Third, I argue that the social interaction is already encoded in the trajectory data without an explicit label. Since the trajectory data contains data of k-agents, a model with sufficient capacity should be able to learn how an ego agent behaves with respect to other k-agents. Not only the labels lack capacity to depict a more complex scenario (imagine, within 5 seconds, a car behind an ego agent overtakes and appears in front in the end; would the label, ‘Decelerate for moving lead vehicle’, mean a leading vehicle in the beginning or in the end?), but also fails to capture relationships with respect to j <= k agents (e.g., a car behind the ego is an ambulance car and an ego slows down for the car behind while keeping its relative speed and position to the leading vehicle). Instead of enumerating intent labels, it may be better for queries to indicate an ego and j-agent trajectories.

Lastly, the size of the dataset is not large. From the content in the paper alone, I cannot be confident that a model can learn social interactions especially when one is significantly different than known ones (i.e., when an intent cannot be characterized as a linear combination of known intents).

**Additional Feedback:**

See above.

**Correctness:**

The submission is missing a number of items in the checklist including report on error bars, details on dataset and all supplementary materials that should accompany the paper. As it currently stands, the submission is incomplete.

**Documentation:**

The dataset is purportedly on GitHub but, at the time of the review, the repository only contained README which contained license information. As per appendix, the dataset is to be maintained and corrected accordingly when an error is reported.

**Ethics:**

I do not see any ethical concerns related to the paper.

**Relation To Prior Work:**

I lack knowledge in the autonomous vehicle domain that I cannot confidently cover all prior contributions. However, amongst the prior work presented by paper, it was not clearly indicated how this work differs from work of Segal et al.

**Summary And Contributions:**

The paper develops a benchmark dataset for trajectory retrieval task in driving scenario.  It introduces a new dataset where a trajectory annotated with intent that is to be matched and ranked against other trajectories based on a similarity metric. To set the baseline, the paper evaluates the dataset against four embedding functions on top of a feedback based method.

---

> ### Author Response · Authors · 2021-07-12
> **Response to Reviewer FmPW**
>
> Thank you for your feedback and suggestions. We address your concerns below, and we've also updated our dataset page.
>
> > "there is no strong evidence in paper text that intent was used in any of the experiments in section 5.1."
>
> Yes, experiments in section 5.1 are our baselines that do not leverage the intent labels. We compare this with experiments in section 5.2 that use the labels with a triplet loss, and experiments in section 5.3 that utilize relevance feedback with respect to the labels.
>
> > "Second, if intent were used meaningfully to contribute to improvements ranking, there should be an ablation study that corroborates its utility."
>
> Experiments in section 5.1 effectively serve as the ablation study you're looking for.
>
> > "social interaction is already encoded in the trajectory data without an explicit label"
>
> Yes, this is the case, but we still find it insufficient for good retrieval in section 5.1. Please let us know if we misunderstood your point here.
>
> > "Lastly, the size of the dataset is not large"
>
> Although relatively small, our dataset has high quality (and expensive) labels and the experimental setup matches the practical use-case. See global comments for more details.
>
> > "it was not clearly indicated how this work differs from work of Segal et al.”
>
> Segal et al.'s work is excellent, and demonstrates widespread interest in and importance of this problem to AV companies. Unfortunately they have not been able to release a public dataset.  Our main contribution is the first ever open benchmark in this setting.  See global comments for more details.

---

### Official Review · Reviewer_EnmT · 2021-07-04
**The Argoverse Trajectory Retrieval Benchmark**

**Rating:** 6
**Confidence:** 4

**Strengths:**

1. The proposed work is the first benchmark of its kind, which contains trajectories annotated with clearly defined intents.
2. The proposed work can facilitate research in trajectory retrieval and intent prediction and can be applied to the field of autonomous driving and beyond.
3. The proposed work can promote explainability in the maneuvering of autonomous vehicles and help to query interesting scenes in large-scale autonomous driving systems.
4. The proposed work builds upon an existing trajectory dataset, which enables multitask learning and unsupervised learning that can leverage the unlabeled data.

**Weaknesses:**

1. Although the proposed dataset is the first of its kind, the number of scenarios is relatively low compared with current deep learning datasets.
2. Experiments can be more sufficient. For example, the role of triplet loss in Section 5.2 is not clear because the results are inconsistent. The authors should include the proposed joint training strategy using triplet loss.
3. The filtering rationale discussed in line 175 is not mentioned in the paper, plus not justified. Also, it seems that <10% of trajectories are selected for labeling. Would it be helpful to release some "distractors" with no clear intent found?
4. The selection of the types of intent is not justified. It would be helpful for the authors to explain how they arrived at the set of 13 intents proposed in the paper.

**Additional Feedback:**

1. For relevance feedback, only m=5 is shown. It might be helpful to show the impact of different numbers/rounds of relevance feedback on the performance. The application of relevance feedback should also be discussed more clearly to motivate this setting.
2. It would be interesting to extend the work to leverage the unlabeled data by unsupervised learning.
3. It would also be interesting to use a trained model to query the unlabeled dataset in Argoverse and showcase the potential of this work for automatic labeling of intents.
4. Since sometimes intents might coincide, it would be interesting if the authors can visualize the correlation between intents.

**Clarity:**

In general, the paper is well-presented.
1. Equation (1) is confusing; NDCG should be explained more clearly.
Misc:
* Line 178: "then THEN decelerate".
* The best numbers in Table 2 can be bolded to make the table easier to read.

**Correctness:**

The benchmark is designed in a fairly straightforward way, and the setup is clearly defined. The benchmark is developed based on the existing Argoverse dataset. The baseline experiments in the dataset include multiple learned and non-parametrized embedding functions. However, there are certain areas that are not clear:
1. The rationale of hand-selection in the dataset (such as selecting the query set) is not explained clearly. Will the hand-selection part introduce undesired bias? In addition, the hand-selection strategy might not be optimal for scalability.
2. The paper mentions that each trajectory is 5 seconds long, but the authors have not mentioned how these trajectories are trimmed, or if the trimming was reused from the Argoverse Motion Forecasting dataset. If the former is the case, what's the rationale for the trimming? If the latter is the case, is the trimming strategy suitable for both of the two benchmarks?

**Documentation:**

In the paper, the authors provided lots of information about the dataset. However, the rationale of hand-selecting queries discussed earlier can be discussed in detail.

**Relation To Prior Work:**

The related work is well-reviewed and presented, and the authors compared the proposed work to prior ones. However, is there existing dataset that provide any form of annotation for intent? If so they should better be included in the paper.

**Summary And Contributions:**

In this work, the authors proposed the Argoverse Trajectory Retrieval Benchmark, a retrieval benchmark that is built upon the Argoverse Motion Forecasting dataset. The goal of this benchmark is to advocate works in finding similar trajectories from a dataset given some query trajectory. The authors split 2,795 trajectories into query and retrieval sets and annotated 13 types of intents with 5 only present in the test set. Baseline experiments on the retrieval task are also shown using nearest-neighbor retrieval with different embedding functions. The authors also showed fine-tuning using contrastive learning and a "relevance feedback" framework that can provide the ground-truth relevance feedback to a small set of initially retrieved trajectories.

---

> ### Author Response · Authors · 2021-07-12
> **Response to Reviewer EnmT**
>
> Thank you for your feedback and suggestions. We address your concerns below, and we've also updated our dataset page.
>
> > "the number of scenarios is relatively low"
>
> Although relatively small, our dataset has high quality (and expensive) labels and the experimental setup matches the practical use-case. See global comments for more details.
>
> > "the role of triplet loss in Section 5.2 is not clear"
>
> Our conclusion is that triplet loss, an objective used to improve learned representations, does not help in our setting (line 274). Note that it is common to fine-tune embeddings using a smaller set of labels after an initial larger-scale unsupervised learning stage.
>
> > "The filtering rationale discussed in line 175 is not mentioned in the paper"
>
> It is common in the AV industry to filter out uninteresting scenarios (e.g. straight line constant speed trajectories). In practice, it is infeasible to search through all trajectory data (analogous situations abound in other retrieval settings such as web search).  Note that in Figure 3c, our dataset does contain trajectories without any intents. We can make this more clear in the paper.
>
> > "The selection of the types of intent is not justified."
>
> Our intents were chosen to capture interesting and rare behaviors, many of which cover common failure cases found in submissions for our Argoverse Motion Forecasting competition.  See global comments for more details.
>
> > "is the trimming strategy (of Argoverse Motion Forecasting dataset) suitable?”
>
> Yes, we found 5s trajectories adequate for demonstrating multiple intents. We did not look into finding the optimal episode length, which is a potentially interesting direction for future work.
>
> Additional Feedback - We thank the reviewer again for their attention to detail and quickly comment on their following suggestions. 1) Our benchmark includes a single round of relevance feedback, but we would expect multiple rounds to improve performance, at the cost of more human intervention. In practice, we find that engineers are willing to spend a few minutes giving feedback in order to obtain better retrievals. 2) WIMP, FEAT, and VNET embeddings were all learned using the entire unlabeled training set (>140,000 scenarios). 3) Yes, automatic labeling of intents would be great, but we found there to be a huge gap between our current best automated approaches and expert annotations. 4) The confusion matrix for intents is shown in Figure 3a.

---

### Official Review · Reviewer_swJ7 · 2021-07-04
**The Argoverse Trajectory Retrieval Benchmark**

**Rating:** 6
**Confidence:** 3
**Clarity:** Yes the paper is clearly written.

**Strengths:**

The paper is clearly written with sufficient initial experiments demonstrating a promising and challenging research area. The dataset, as well as the proposed task, opens a new area that allows future AV research to model driving behaviors and to analyze the ambiguous driving intent.

**Weaknesses:**

The scale of the dataset seems to be a concern in my opinion. With a total number of 2795 trajectories, I have some doubts that the current status of the dataset would be sufficient to train large-scale models. I hope to see a larger number of trajectories in future iterations.

Another concern is that it seems that there are many types of intents that are related to "Deceleration", at least in the training set. I am concerned that this introduces some intrinsic bias in the dataset. Was there any reason why most of the intents are related to "Deceleration"? Do you plan to mitigate this potential bias?

**Additional Feedback:**

N/A

**Correctness:**

Yes the dataset is constructed in a sound way, but there are some concerns as stated in the "Weakness" section. The evaluation criteria and experiments seem reasonable.

**Documentation:**

Yes the dataset details are listed in the appendices. It should be noted that the dataset github page only contains a README currently.

**Relation To Prior Work:**

Yes this is stated in the related work.

**Summary And Contributions:**

This paper proposes an interesting dataset that focuses on the new task of trajectory retrieval, named the Argoverse Trajectory Retrieval Benchmark. The dataset proposed contains 2795 trajectories of 13 different driving intents that are annotated by human annotators as well as some domain experts. It is a rather challenging task and the experiments show a relatively large performance gap between current methods and the upper bound.

---

> ### Author Response · Authors · 2021-07-12
> **Response to Reviewer swJ7**
>
> Thank you for your feedback and suggestions. We address your concerns below, and we've also updated our dataset page.
>
> > "The scale of the dataset seems to be a concern in my opinion."
>
> Although relatively small, our dataset has high quality (and expensive) labels and the experimental setup matches the practical use-case. See general response for more details.
>
> > "Was there any reason why most of the intents are related to "Deceleration"? Do you plan to mitigate this potential bias?"
>
> Deceleration events happen to be the most causally interesting. For example, an AV may comfortably veer around an actor decelerating to turn, but should exhibit more caution if the actor is decelerating for another (potentially unobserved) actor. In general, our intents were chosen to capture interesting and rare behaviors of interest to industry engineers, many of which cover common failure cases found in submissions for our Argoverse Motion Forecasting competition. We aim to mitigate the potential bias with held-out test intents. See global comments for more details.

---

### Author Response · Authors · 2021-07-12
**Global comments to all reviewers**

We thank all reviewers for their feedback and suggestions. Here, we address two common concerns raised by all reviewers and summarize our contributions. We respond to each reviewer individually below. Please note that we have also updated our dataset page to include the training data as well as the code used in our experiments.

### Our contributions

To the best of our knowledge, we have released the first ever open benchmark for multi-intent trajectory retrieval for traffic scenes.  This benchmark was carefully constructed to reflect scenarios that are relevant to modern use cases such as autonomous driving.  The dataset curation and annotation were performed by domain experts, and so we feel that the quality of data is very high. The timeliness is also evident from recent papers (e.g., Segal et al.) studying this retrieval setting but not publishing any open benchmarks.  We thus feel that our contribution is a significant advancement, by helping to establish rigorous benchmarking protocols for multi-intent trajectory retrieval, which will spur future research in this area (analogous to how the Caltech 101 dataset eventually led to ImageNet).

### 1) Size of dataset

One common concern is that the dataset may be too small.  We believe that the dataset is still rich enough to serve as an initial benchmark in this area of research, due the following points:
- Although data is plentiful in the AV domain, only a small percentage actually contain traffic scenes with interesting behaviors. Our initial filtering step uses labeling functions already used in the industry to filter out uninteresting scenes (e.g. straight line trajectory with constant speed). This is analogous to other retrieval domains such as web search, where the list of candidate results for search query is a tiny subset of the vast number of web pages on the World Wide Web.
- It is now common-place to pre-train embeddings via self-supervised and unsupervised techniques, and then adapt to downstream tasks using a modest amount of high-quality supervised training data.  Our experiments follow that protocol by using embeddings trained on large unlabeled trajectory datasets.  In establishing this workflow for multi-intent trajectory retrieval, a major design decision is how best to create a (modestly sized) high-quality supervised training set, which is a key contribution of our work.
- The collection of high quality training data requires extensive effort from (very expensive) domain experts.  Trajectories were labeled by domain experts who have spent considerable amounts of time understanding and becoming familiar with the data. Our dataset is only a  starting point and we envision future iterations to contain more labels as we streamline this process.

### 2) Bias with intent selection

A second common concern is that our hand-picked intents set introduces an intrinsic bias that we do not account for.  In actuality, this bias is very much intentional (which we will make more clear in the paper).  Our goal is to focus on practically relevant scenarios for autonomous vehicles (AVs), which translates to focusing on trajectories with interesting and rare behaviors.  Our intents cover common failure cases of submissions for the Argoverse Motion Forecasting competition and reflect search intents provided by (domain expert) AV engineers. By analogy, useful datasets in medical domains are similarly biased toward high-risk scenarios (rather than, say, collecting CT scans on a representative sample of the entire population). Lastly, we highlight that we try to prevent overfitting to our intents by introducing not only a test set of queries, but also a test set of intents as well.

---

### Decision · Program_Chairs · 2021-07-27

**Decision:**

Reject

**Comment:**

Through discussion with the other AC and PC, we conclude that the paper is still lacking in experiments and justification for the limited size and variety of the proposed dataset. The overall rating of the paper is leaning towards rejection and we don't see a strong argument to override the reviews.